# Processed Meat Characteristics between Commercial Duroc-Sired and Heritage Breed Large Black Pigs

**DOI:** 10.3390/foods11152310

**Published:** 2022-08-02

**Authors:** Yufei Guo, Katharine Sharp, Harvey Blackburn, Brian Richert, Kara Stewart, Stacy M. S. Zuelly

**Affiliations:** 1Department of Animal Science, Purdue University, West Lafayette, IN 47906, USA; guo335@purdue.edu (Y.G.); sharp78@purdue.edu (K.S.); brichert@purdue.edu (B.R.); krstewart@purdue.edu (K.S.); 2United State Department of Agriculture—Agriculture Resource Service, Washington, DC 20250, USA; harvey.blackburn@usda.gov

**Keywords:** pork, processed meat, Duroc, large black, heritage breed pork, pork quality

## Abstract

This study examined the meat processing characteristics between Duroc-sired (DS) and heritage breed Large Black (LB) pigs fed high forage (FIB) or commercial diets (CON). Fifty pigs (DS, *n* = 25; LB, *n* = 25) were used in a 2 × 2 factorial design with breed and diet as fixed effects. Processing traits were measured for fresh bellies, bacon slices, and sausage patties. Results showed that DS bellies were longer (*p* < 0.001) but thinner (*p* = 0.026). CON bellies were longer (*p* = 0.005) and thicker (*p* < 0.001). LB bellies had decreased lean area (*p* < 0.001) and processing yield (*p* = 0.001). DS patties were lighter (*L**, *p* < 0.001) and less red (*a**, *p* < 0.001). LB had increased belly firmness (*p* = 0.053). The CON diet resulted in better particle definition (*p* = 0.010) in patties as well as a larger slice area (*p* = 0.048) in bacon slices. A breed × diet interaction was observed for lipid oxidation (*p* = 0.001). The findings provided novel insight into the comparison between these breeds and diets. While LB pork may have niche market value, the integration of this breed into commercial bacon processing has limitations in composition that need to be further evaluated to improve the product desirability.

## 1. Introduction

Pork quality is the result of a combination of genetic and environmental factors that impact meat color, marbling, water-holding capacity, and tenderness. Processed meat quality requires the incorporation of functional ingredients, which alter biochemical characteristics, and mechanical manipulation, which adds the need for firmness for production purposes. Pork from commercially produced genetic sources is readily available for public purchase, however, based on customer experiences, some pork quality shortcomings have been detected [1]. Schwab et al. [2] found when comparing pork from different generational eras that more recent genetics had pork with less pork flavor and more off-flavor. Further works have shown older genetic lines to produce pork with a less bland flavor than commercial pork from commercially produced pigs [3]. Schwab et al. [2] suggested that selection for carcass composition over time had come at the expense of palatability traits. Thus, studies exploring various breeds and nutritional management strategies could be helpful in resolving the challenge faced by commercial swine production.

Some consumers have started to broaden their searches for the “nostalgic taste”, and heritage breed pork has started to form a niche market to offer these consumers an alternative to commercially produced pork [4]. Large Black pigs, as one of the pasture-raised minor swine breeds in the United States, have not been through intense genetic selection for increased carcass leanness. The breed has maintained the “old time” characteristics, when fatty pigs were desirable due to lard demand and were marketed for superior meat quality. Large Black pigs have historically been raised under outdoor management systems, and due to their grazing nature, the Large Black pigs consume a diet that contains a greater amount of fibrous material and lower energy, unlike commercial swine feed. Although the Large Black pigs appear to be well fitted for the niche market, relatively limited data are available regarding the breed’s carcass and processing characteristics.

Whitley et al. [5] examined pork quality and sensory characteristics in Large Black × Yorkshire (LBY) crosses in comparison to other Yorkshire crosses; thicker backfat and smaller loin muscle areas were observed in LBY pigs. Aside from findings by Whitley et al. [5] and Wang et al. [6], there has been a very limited number of studies evaluating purebred Large Black fresh pork quality, and no previous work to evaluate processed meat quality. Furthermore, the commercial Yorkshire breed has not focused on pork quality in selection criteria, similar to the Duroc breed. Due to diet, genetics, and methods of husbandry variabilities, a meaningful comparison must include both breed and diet variables. This study was designed to fill this gap in knowledge by comparing Duroc and Large Black genetic lines fed high forage or commercial diets to determine differences regarding pork processing characteristics. Therefore, the objective of this study was to examine differences in processing characteristics between Duroc-sired and Large Black genetic lines with high forage and commercial diets.

## 2. Materials and Methods

All animal use and procedures were approved by the Purdue Animal Care and Use Committee (1801001688).

### 2.1. Animal Harvest

Fifty pigs were utilized in the study with a 2 × 2 factorial design of breed and diet: Duroc-sired (DS, *n* = 25 pigs) and Large Black-sired (LB, *n* = 25 pigs). All the pigs were weighed and allocated to two dietary treatments: Fiber (FIB) or Control (CON), (LB FIB, *n* = 14; LB CON, *n* = 11; DS FIB, *n* = 14; DS CON, *n* = 11). Dietary treatments were fed throughout the grow-finish period in six phases (Appendix A Table A1). The CON diet was concentrate-based and included corn, soybean meal, and distiller’s dried grains with solubles (DDGS). The FIB diet contained wheat middlings (1–10%) and dehydrated alfalfa meal (7.5–20%) in replacement of corn and soybean meal to mimic the more natural diet of the LB considering their pasture-raised and foraging nature. Pigs were slaughtered at a common age with variations in BW among genetics (DS 125 ± 2.23 kg, LB 99 ± 2.28 kg) at Purdue University after the completion of diet phase 6.

Electrical stunning was applied to each pig before exsanguination during slaughter. Dehairing was achieved using a scalder, therefore no skin or backfat was removed during the process. Hot carcass weight was recorded (Rice Lake Weighing Systems, Rice Lake, WI, USA), and carcasses were chilled for 24 h in a blast cooler (4 °C) prior to further processing. At one day post mortem, the right side of each carcass was fabricated into wholesale cuts with picnic shoulders, Boston butts, and boneless, skin-on bellies, which were utilized for this study.

### 2.2. Boneless Belly

All bones and cartilages were removed from the pork bellies according to the guidelines stated in the Institutional Meat Purchase Specifications for item No. 408—Pork Belly [7], and fresh belly weight was recorded. For belly thickness and length, each belly was laid skin side down. The length of the belly was measured from the anterior (blade) end to the posterior (flank) end (Figure 1). Belly thickness was measured using data collected from 4 locations at points 25% and 75% of the distance from the blade end (Figure 2). A sharp knife was used to make small perpendicular cuts on the belly to mark the locations and a ruler was used to measure the thickness at each location. The final thickness of the belly was calculated by averaging the measurements collected at all 4 locations. Belly firmness was measured similar to methods outlined by Rentfrow et al. [8], by suspending the belly (skin side down) horizontally over a PVC pipe (diameter = 8.89 cm). The PVC pipe was attached to a board with an X-axis and Y-axis. The coordinates were used to measure the distance between Point A (Blade end) and Point B (Flank end) of the belly. A smaller distance between Point A and Point B suggested decreased firmness and a greater distance between the points indicated that it was a firmer belly.

### 2.3. Sausage

An individual batch of sausages was made from each carcass using trim from the picnic shoulder and Boston butt. Lean and fat trim were collected to generate an 80:20 lean-to-fat ratio for each batch. Each batch was ground twice using a tabletop grinder with a #22 2.4 mm plate. A paddle mixer was then used to mix the ground pork and the seasoning (0.023 kg of Pork Sausage Seasoning Blend 10 from A.C. Legg, INC, Calera, AL, USA) for a standardized time of mixing. Five 0.14 kg sausage patties from each batch were made using patty molds and were randomly assigned to a display time: 0 d (1 patty), 3 d (2 patties), or 7 d (2 patties). Each observation day’s patties were packaged on a foam tray, PVC overwrapped, and placed under retail display lighting to evaluate color changes and lipid oxidation over time.

### 2.4. Color

Color measurements of commission internationale de l’éclairage (CIE) *L**, *a**, and *b** values were obtained using a Minolta CR-400 Chroma Meter (Konica Minolta, Chiyoda, Tokyo, Japan). Minolta CR-400 (8 mm aperture, 2° observer) was calibrated with PVC wrap on the eye and the measurements were taken using illuminant D65 through the PVC packaging. On each display day, the surface color of each patty within a package (0 d = 1 patty, 3 d = 2 patties, 7 d = 2 patties) was measured in triplicate. After color measurements were taken on their respective days, all patties were vacuum packaged individually and placed in a −40 °C freezer for lipid oxidation analysis.

### 2.5. Fat Smear

A fat smear scale (Figure 3) was made to assess the fat smearing condition of each sausage patty. The scale ranges from 1 (excessive fat smearing) to 8 (very little fat smearing). A score of 8 showed fat particles that held their shape and the edges of the particles were easily identified. A score of 1 suggests that the fat particles were smeared and mixed with the lean part of the patty, and the edges of the fat particles were unclear to identify. Both patties for each displayed day were scored, and an average was taken to calculate the final fat smear score for each batch.

### 2.6. Lipid Oxidation

Lipid oxidation for the sausage patties was examined with the 2-Thiobarbituric acid reactive substances assay (TBARS). The TBARS protocol was modified from Buege and Aust [9]. One (1) patty from each displayed package (0 d, 3 d, and 7 d) was taken to perform TBARS, with duplicate samples taken from each patty. All solutions (butylated hydroxyanisole (BHA) and trichloroacetic acid (TCA)) were made prior to each analysis day to ensure consistency throughout the analysis. Absorbance was read at 531 nm of the supernatant using a 96-well plate.

### 2.7. Bacon

Fresh bellies were weighed to obtain fresh weight. Bacon Brine Additive with Salt from A.C. Legg. INC. (Blend JM-95-145-000, Blended of Salt, Sodium Phosphates (10.47%), Sodium Erythorbate) was mixed with water to make the brine solution. Brine was injected with a manual compressed air stitch pump until the belly reached 110% of its fresh weight, and the pumped weight was recorded. Pump uptake (%) was calculated with the equation: (pumped weight—fresh weight)/fresh weight × 100. The pumped bellies were rested for 3.5 h before placing into the smokehouse. Bellies were thermally processed for 4 h until the internal temperature reached 62 °C. Thermally processed bellies were cooled to 1 °C internal temperature and weighed to obtain the cooked weight. The processing yield (%) was calculated with the equation: (cooked weight/fresh weight) × 100. Bellies were then cut at 25%, 50%, and 75% of the distance from the blade end, and a 0.64 cm thick slice was taken from each cut surface for visual image analysis.

### 2.8. Visual Image Analysis

Each 0.64 cm bacon slice removed at 25%, 50%, and 75% of the distance from the blade end of the thermally processed bellies was labeled and photographed with a standardized ruler for scale reference using the modified protocols of Scramlin et al. [10]. Each bacon slice was photographed at the same distance from the camera. Images of each slice were analyzed for slice length (SL; cm), slice area (SA; cm^2^), and slice lean area (LA; %) with Adobe Photoshop (22.2.0 Release). The ruler tool was used to measure the number of pixels in 1 cm per image as well as the slice length in pixels. The magnetic lasso tool was used to trace the outline of each slice and the area selected was automatically provided by the software for SA. The same technique was used to measure the lean muscle area and the lean % was calculated with the equation: (lean area/slice area) × 100. All measurements were recorded in pixels and then converted to cm (SL) or cm^2^ (SA). All bacon measurements, data obtained from 25%, 50%, and 75% of the thermally processed belly, were averaged together to calculate the SL, SA, and LA for each carcass.

### 2.9. Statistical Analysis

The data were analyzed as a 2 × 2 factorial design with breed and diet as fixed effects. All the data were analyzed with RStudio (1.2.1335) with least square means determined significant *p* < 0.05. Functions used in the analysis included linear model and ANOVA. Tukey’s test was performed if any interactions such as breed × diet or measurements × days were found to be significant.

## 3. Results

### 3.1. Physical Belly Characteristics

The main effects of breed and animal diet fresh belly and bacon characteristics are reported in Table 1 and Table 2. Both breed (*p* < 0.001) and diet (*p* = 0.001) were significant for fresh belly weight and no breed × diet interaction (*p* = 0.5571) was found. Fresh bellies from DS pigs were 0.86 kg heavier than bellies from LB pigs. Pigs fed the CON diet had heavier bellies than pigs fed the FIB diet by 0.76 kg. Data showed significance in breed (*p* < 0.001) and diet (*p* = 0.005) for belly length, but no breed × diet interaction (*p* = 0.724) was observed. LB bellies were 4.7 cm shorter than DS bellies, and CON bellies were 3.3 cm longer than FIB bellies. Breed (*p* = 0.026) and diet (*p* < 0.001) were found to be significant for average belly thickness, and no breed × diet interaction (*p* = 0.5230) was discovered. Bellies from DS pigs were 0.36 cm thinner than the LB bellies, and CON bellies were 0.66 cm thicker than FIB bellies. Belly firmness was measured by the distance between the blade and flank end of each belly when suspended over a PVC pipe. Although both breed (*p* < 0.001) and diet (*p* < 0.001) were shown to be significant for belly firmness, a very strong trend for a breed × diet interaction (*p* = 0.053) was observed (Figure 4). LB CON pigs had the firmest bellies and DS FIB had the softest bellies, while LB FIB and DS CON were intermediate. No breed (*p* = 0.139) or breed × diet interactions were (*p* = 0.197) observed for belly pump uptake. However, a diet effect was approaching significance (*p* = 0.053) for brine pump uptake, as FIB had a slightly greater pump uptake than CON bellies by 0.41%. Breed (*p* = 0.001) was found to be significant for belly processing yield, as DS bellies had 0.95% more belly processing yield than LB bellies, while no diet (*p* = 0.439) effect and breed × diet interaction (*p* = 0.273) were observed.

### 3.2. Bacon Quality and Characteristics

Bacon quality results are reported in Table 2. Breed did not have an effect on bacon SA (*p* = 0.676) or SL (*p* = 0.065). However, DS pigs had 20.32% more LA (*p* < 0.001) than bacon slices obtained from LB pigs. Contrary to breed, diet had a significant effect on bacon SA (*p* = 0.048), as bacon slices from CON bellies had a greater SA than FIB bellies. Diet did not have any effect on total slice length (*p* = 0.645) nor lean area (*p* = 0.203). No breed × diet interactions were found for any bacon slice characteristics (*p* > 0.05).

### 3.3. Sausage Characteristics

Sausage patties made from the picnic shoulder of each pig were examined for fat smearing, with results shown in Table 2. No breed effect (*p* = 0.398) or breed × diet interaction (*p* = 0.302) was observed for fat smearing; however, diet (*p* = 0.010) was significant. Patties made from CON pigs had greater fat smearing scores compared to FIB pigs, which indicated that CON patties had the best particle definition and least amount of fat smearing.

Time under retail display lights negatively impacted the color and lipid oxidation of the sausage patties (Figure 5 and Figure 6). Display time was found to be significant for *L** (*p* = 0.005), *a** (*p* < 0.001), and *b** (*p* < 0.001), which was expected given that meat product color typically changes over time. TBARS value increased during the retail display period, which indicates that more lipid oxidation occurred as time went on, which was also expected. No breed × diet × days interactions were found in *L**, *a**, *b*,* or TBARS measurements. Breed was significant for *L** (*p* < 0.001) and *a** (*p* < 0.001) but did not impact *b** over the entire display period (Figure 5). DS patties had a greater *L** value but lesser *a** value when compared to LB pigs, therefore patties from LB pigs were darker and redder than those from DS pigs. There was no diet effect or breed × diet interaction found for any of the instrument color measurements. Regardless of time, a breed × diet interaction (*p* = 0.001) was observed for patty lipid oxidation, as DS CON patties had the least amount of lipid oxidation regardless of retail display time, whereas LB CON patties had the most lipid oxidation (Figure 6). 

## 4. Discussion

The objective of this study was to determine differences in pork processing characteristics between commercial Duroc-sired genetics and Large Black genetic lines fed diets similar to the commercial management of their respective breeds. There are relatively few studies that have examined the processing traits of Large Black pigs, and none that have evaluated processed meat characteristics. Results from multiple studies suggested that the Duroc genetics resulted in superior carcass qualities compared to other breeds, such as increased intramuscular fat and loineye area and decreased subcutaneous fat [11,12]. In the present study, the DS pigs were used to represent pigs from modern-day swine production since the Duroc genetics are commonly used in commercial settings due to their ability to produce high-quality offspring. Contrary to the commercial production pigs, the LB pigs represented products from the growing heritage breed niche market, where the breed has undergone very minimal genetic selection for lean percentage. Two types of diets respective to breeds (CON for DS, FIB for LB) were used in the study since DS and LB pigs are typically produced under different management systems. The CON diet, which contained corn and soybean meal, was composed to represent the common feed utilized in commercial swine settings. The FIB diet was high in alfalfa to mimic the natural foraging diet of LB pigs since the breed tends to be pasture-raised.

The differences in fresh belly characteristics found between breeds could be attributed to LB pigs’ smaller sizes compared to DS pigs at the time of slaughter. This could be indicative of the lower production efficiency of LB pigs compared to DS pigs. Due to DS pigs’ larger sizes and higher lean percentage from genetic selection over time, it was expected that the DS pigs would have heavier and longer bellies than LB pigs, which could result in greater bacon slice yields. Regardless of breed, pigs fed the CON diet had heavier and longer bellies than pigs fed the FIB diet, this is probably due to a higher energy concentration in the corn and soybean meal-based diet. LB pigs had a visibly greater amount of backfat, which was consistent with previous findings by Whitley et al. [5]. The greater amount of subcutaneous fat contributed to increased belly thickness as well as belly firmness in LB pigs, which can be advantageous for bacon slicing yields [10]; however, excessive fat may reduce consumer appeal. The breed × diet interaction observed in belly firmness suggested that LB CON pigs had the firmest bellies whereas DS FIB pigs had the softest bellies among all treatments. It was interesting to note that feeding the naturally opposite diets (LB fed CON and DS fed FIB) resulted in the firmest belly (LB CON) and the softest belly (DS FIB). This could be due to differences in metabolisms between the breeds and how their bodies interacted with the diets. Future fatty acid analysis could be helpful to examine the adipose composition and determine if the diets had an impact on fat saturation. Although all the bellies had a 10% target brine uptake, FIB bellies displayed a trend for better brine absorption based on the tendency observed in diet effect on pump uptake, which could result in improved cured meat color. Previous literature has shown that bacon slices with a larger lean area had greater processing yield [10]. This could be attributed to the fact that a larger lean area means there would be more protein present to absorb brine. This was replicated in the current study, where bacon slices from DS pigs showed 20.3% more lean area and 0.95% more processing yield than LB pigs. Although there was a difference in pump uptake, it is not believed that the sensory attributes of the bacon would be affected by the variations in brine uptake [13].

The impact of diet on sausage patty fat smearing could be attributed to fat saturation, as FIB patties had the most fat smearing. Saturated fatty acids have a higher melting point, which tends to lead to firmer fats with improved ability to maintain structural integrity throughout processing techniques such as grinding and stuffing. Increased fat smearing could be indicative of increased unsaturated fatty acid in pigs fed the FIB diet. Although the breed was not significant for patty fat smearing, differences in fatty acid composition due to genetics is still anticipated. The lack of significance in breed’s effect on fat smearing results could be due to diet having a greater influence on fat composition than genetics.

Lipid oxidation negatively impacts food quality [14] since it can cause rancidity which is detrimental to product flavor. The patties in the study were packaged with PVC wrapping, an aerobic packaging method, and placed under retail lighting. With light and oxygen being pro-oxidants, it was expected to observe increased lipid oxidation with increased retail display time. Regardless of time, a breed × diet interaction was observed in TBARS values. DS CON patties had the least amount of lipid oxidation, which was the most desirable. However, LB CON patties resulted in the most lipid oxidation. This was unexpected, as previous work comparing Large Blacks to commercially sired pigs found no differences in the total amount of unsaturated fats [6] that could result in differences in lipid oxidation. It is possible the Duroc genetic line has been genetically selected to suit modern commercial swine production and adapted to the high-energy diet. However, it was likely that LB pigs were not able to effectively metabolize the corn and soybean meal-based diet since, historically, the breed has been raised outdoors with a high fiber diet due to its grazing nature. The additional analysis of fatty acids may provide insight into this variation, and could merit investigation in future work.

The impact of retail display time on sausage patty color measurements (*L**, *a**, and *b**) was expected since color deteriorates over time. The present study found that LB patties were darker and redder than DS patties even though they were raised in the same confined environment during the project. It is speculated that there could be variations in muscle fiber types due to genetics among the breeds which resulted in the color differences. Muscle fiber typing could potentially be performed for future studies to further examine the physiological differences between DS and LB genetic lines.

The further analysis of sensory characteristics is warranted to determine if differences found within the analytical data would translate to human subjects. Sensory analysis was intended in this work; however, institutional restrictions related to the pandemic prohibited sensory analysis from completion.

## 5. Conclusions

The study provided various novel insights into the processing characteristics of LB and DS pigs fed CON and FIB diets. In terms of breed, the DS pigs had a greater belly processing yield and contained much higher lean percentages compared to the LB pigs. Although firmer bellies could be ideal during slicing, the excessive amount of backfat on the LB bellies may have outweighed the benefit of increased firmness. Overall, both breeds performed best when fed diets similar to their commercial management practices (i.e., high energy for DS and high fiber for LB). The LB genetics sustained the historic imprint of a time when society favored fat pigs for lard production. Even though the niche market of heritage breed pork continues to grow due to consumers seeking diversity and preferring a historic flavor in commercial pork, there still are limitations to market incorporation of the LB breed. In the future, trained sensory panels, as well as consumer panels, should be conducted to examine any differences in palatability in the breeds and diets of various fresh and processed products. Further studies examining the fatty acid composition of the various fat depots may be insightful in examining differences in animal metabolism. Finally, methods to improve lean percentage and processing yields while preserving potential desirable flavor traits in the LB pigs would be beneficial to improve product quality and probable commercial integration.

## Figures and Tables

**Figure 1 foods-11-02310-f001:**
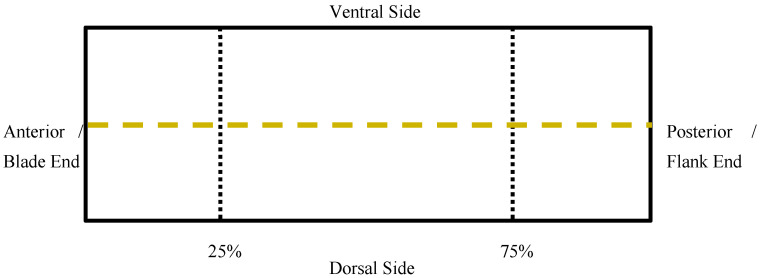
Diagram of belly length measured from anterior (blade) end and posterior (flank) end along the belly midline.

**Figure 2 foods-11-02310-f002:**
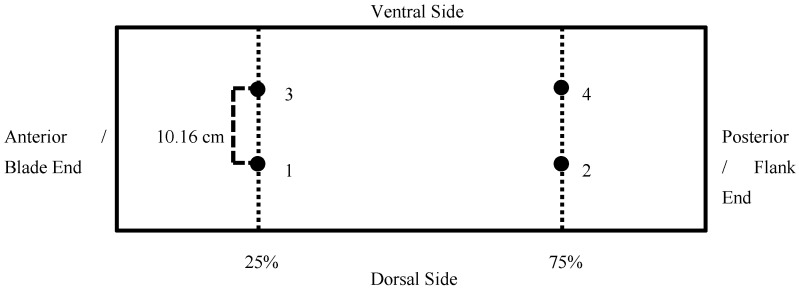
Diagram of belly thickness measured at four locations (1–4) at points 25 and 75% of the length from the anterior (blade) end, spaced 10.16 cm apart.

**Figure 3 foods-11-02310-f003:**
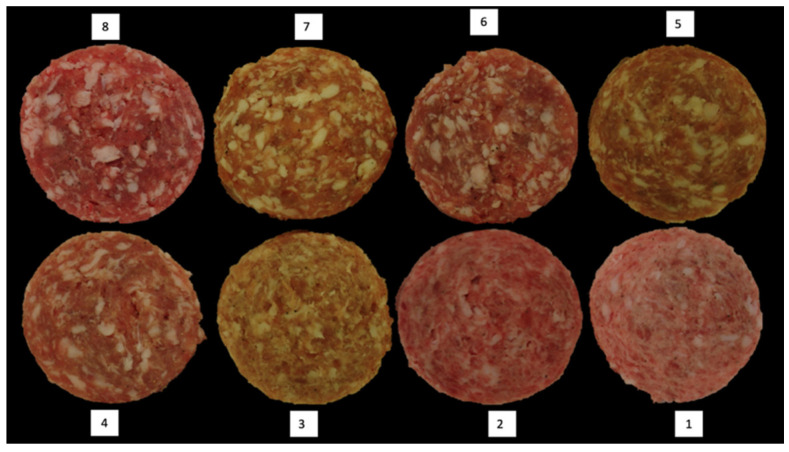
Sausage fat smearing score used to score each sausage patty, with 1 displaying excessive fat smearing and 8 displaying very little fat smearing.

**Figure 4 foods-11-02310-f004:**
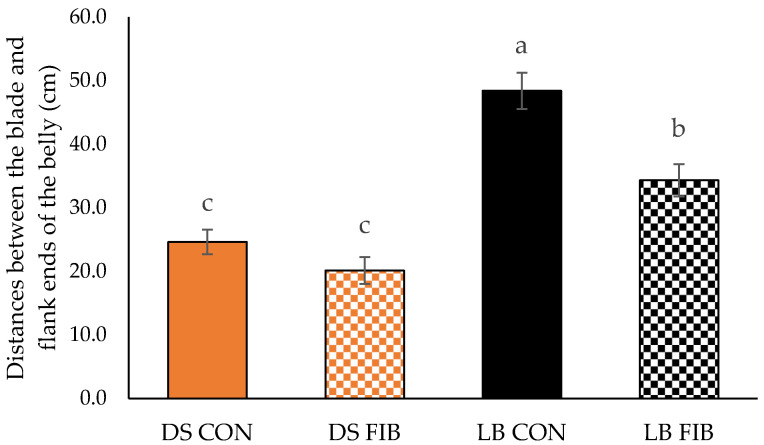
Effect of breed × diet interaction on belly firmness by the distance between the blade and flank ends of the belly placed skin-side down on a stationary PVC pipe (diameter = 8.89 cm). CON = diet of corn–soybean meal, DDGS-based; FIB = diet using increasing amounts of wheat middlings (1–10%) and dehydrated alfalfa meal (7.5–20%) to replace corn and soybean meal in the CON diet. DS = Duroc-sired and LB = Large Black pigs. ^a,b,c^ Means with different superscripts differ (*p* = 0.053).

**Figure 5 foods-11-02310-f005:**
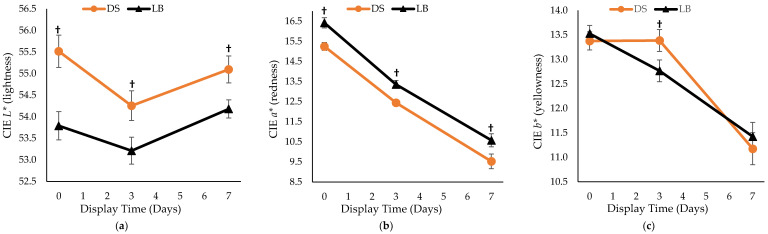
Effect of breed on D65 instrumental color attributes: (**a**) CIE *L** (lightness); (**b**) CIE *a** (redness); (**c**) CIE *b** (yellowness). DS = Duroc-sired and LB = Large Black pigs. ^†^ Means of similar display time lacking the superscript differ due to breed (*p* < 0.05).

**Figure 6 foods-11-02310-f006:**
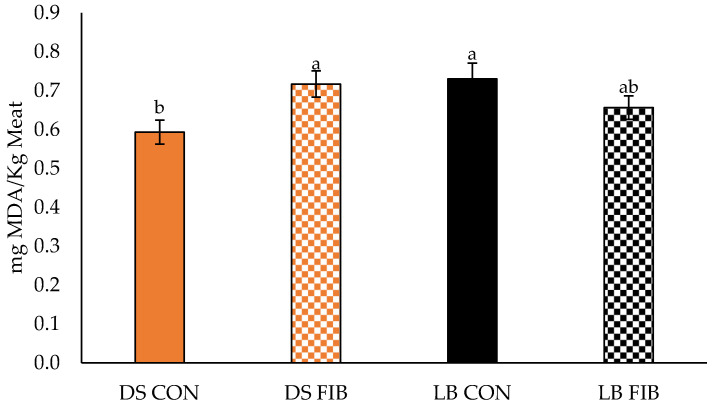
Effect of breed × diet interaction for the lipid oxidation (TBARS value) of sausage patties. DS = Duroc-sired and LB = Large Black pigs. CON = diet of corn–soybean meal, DDGS-based; FIB = diet using increasing amounts of wheat middlings (1–10%) and dehydrated alfalfa meal (7.5–20%) to replace corn and soybean meal in the CON diet. DS = Duroc sired and LB = Large Black pigs. ^ab^ Means lacking a common superscript within the main effect of breed or diet differ (*p* < 0.05).

**Table 1 foods-11-02310-t001:** Effect of breed and diet on fresh belly characteristics.

	Breed ^1^				Diet ^2^			
Belly Characteristics	DS	LB	SEM	Significance of *p*-Value	CON	FIB	SEM	Significance of *p*-Value
Fresh belly wt (kg)	6.64 ^a^	5.78 ^b^	0.24	<0.001	6.64 ^a^	5.88 ^b^	0.24	0.001
Belly length (cm)	64.07 ^a^	59.34 ^b^	1.17	<0.001	63.54 ^a^	60.26 ^b^	1.30	0.005
Belly thickness (cm)	3.72 ^a^	4.08 ^b^	0.18	0.026	4.27 ^a^	3.61 ^b^	0.17	<0.001
Belly firmness (cm) ^3^	22.08 ^a^	40.49 ^b^	0.78	<0.001	36.48 ^a^	27.20 ^b^	3.75	<0.001
Pump uptake (%) ^4^	10.10	9.80	0.21	0.139	9.72	10.13	0.21	0.053
Processing yield (%) ^5^	90.62 ^a^	89.67 ^b^	0.28	0.001	90.27	90.05	0.31	0.439

^1^ DS = Duroc-sired and LB = Large Black pigs. ^2^ CON = diet of corn-soybean meal-DDGS based and FIB = diet using increasing amounts of wheat middlings (1–10%) and dehydrated alfalfa meal (7.5–20%) to replace corn and soybean meal in the CON diet. ^3^ Belly firmness = distance between the blade and flank ends of the belly placed skin-side down on a stationary PVC pipe (diameter = 8.89 cm). ^4^ Pump uptake: (pumped belly weight—fresh belly weight)/fresh belly weight × 100. ^5^ Processing yield: (cooked belly weight/fresh belly weight) × 100. ^a,b^ Means lacking a common superscript within the main effect of breed or diet differ (*p* < 0.05).

**Table 2 foods-11-02310-t002:** Effect of breed and diet on the physical characteristics of bacon slices and fresh sausage.

	Breed ^1^				Diet ^2^			
Processing Characteristics	DS	LB	SEM	Significance of *p*-Value	CON	FIB	SEM	Significance of *p*-Value
SA (cm^2^) ^3^	98.36	96.54	4.42	0.676	102.39 ^a^	93.56 ^b^	4.21	0.048
Total slice length (cm) ^4^	24.37	23.12	0.66	0.065	23.58	23.89	0.67	0.645
Lean area (%) ^5^	39.67 ^a^	19.35 ^b^	1.13	<0.001	28.63	30.20	3.12	0.203
Fat smearing score ^6^	3.56	3.32	0.21	0.398	3.86 ^a^	3.11 ^b^	0.16	0.010

^1^ DS = Duroc-sired and LB = Large Black pigs. ^2^ CON = diet of corn-soybean meal-DDGS based and FIB = diet using increasing amounts of wheat middlings (1–10%) and dehydrated alfalfa meal (7.5–20%) to replace corn and soybean meal in the CON diet. ^3^ SA = slice area. ^4^ SL = slice length. ^5^ LA = (lean area of bacon slice/slice area) × 100. ^6^ Fat smearing score: 1 = excessive fat smearing, 8 = very little fat smearing. ^a,b^ Means lacking a common superscript within the main effect of breed or diet differ (*p* < 0.05).

## Data Availability

The data presented in this study are available on request from the corresponding author.

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
