# Peer review of "Processed Meat Characteristics between Commercial Duroc-Sired and Heritage Breed Large Black Pigs"

_foods, 2022, doi:10.3390/foods11152310_

Round 1

Reviewer 1 Report

This study exploring various breeds and nutritional management strategies in pigs processing. This theme is very interesting for meat processors, farmers, and also meat scientists.

But, the deep revision of this manuscript is necessary.

In the section of Introduction, please add some more information regarding the effect of genetics, breed, diets of pigs on meat quality attributes (color, percentage of the PSE add DFD meat quality).

Sensory attributes are the main quality attributes for meat. Did you perform sensory analysis in this study?

In the sector Discussion, please add some more citations from novel literature data.

In this section, please, reformulate the section regarding Lipid oxidation.

Minor mistake: Please, use L*, a* and b* value in italic.

Author Response

Please see the attachment and the updated manuscript

Reviewer 2 Report

This paper mainly investigated the processed meat, fresh bellies, bacon slices, and sausage patties, as influenced by two pig breed under high forage and commercial diet. The topic is interesting and the result has practical implications. However, such experimental is not strictly designed and well-performed. Three meat products are simply assayed while the parameters concerning edible meat quality, nutritional quality and etc are not extensively and comprehensively determined. Taking sausage as an example, only instrumental color and lipid oxidation are detected, how about the meat texture, sensory evaluation, microstructure and the biochemical factors of pig meat caused by breed and feed in influencing the difference of prepared sausage patties. This paper is mostly like a news or report and is not suitable published as a research article. In addition, some important trait of rearing practice such as feed intake, body weight gain and the ingredient and nutrient composition of feeding diet are lacked. 

Author Response

(The authors gave the same response as above.)

Reviewer 3 Report

1.  Line 63-65: “All the pigs were 63 weighed and allocated to two dietary treatments: Fiber (FIB) or Control (CON), (LB FIB, n=14; LB CON, n=11; DS FIB, n=14; DS CON, n=11).” Why didn't the authors keep the number of repetitions consistent across treatment groups?

2.  Line 65-66: “Dietary treatments were fed through-65 out the grow-finish period (101 or 140 d) in six phases.” Slaughter time varies greatly between the two breeds, and it is well known that the quality of pork can also be affected by the length of feeding time. Therefore, is it meaningful to compare the quality of pork under different feeding time?

3.  Are the nutrient levels of CON diet and FIB diet consistent? The author had better list the composition and nutrient level of both diets.

4.  Figure 3 is not necessary in the paper.

5.   Line 241-243, the authors sat that “Regardless of time, a breed x diet interaction (p = 0.001) was observed for patty lipid oxi-241 dation as DS CON patties had the least amount of lipid oxidation regardless of retail dis-242 play time whereas LB CON patties had the most lipid oxidation (Figure 7).” however, we do not see any significant differences in Figure 7.

6.  There are too few references, so I suggest that this paper refer to more recent studies.

Author Response

(The authors gave the same response as above.)

Reviewer 4 Report

1. The data was previously published in the form of thesis

2. Line 28-29 what the authors meant?

3. section 2.3 please specify how many patties was measured in each group on each storage days. Preferably giving n number

4. Section 2.6. similar to previous one. How many patties were analysed in each group on each storage day. Please clarify.

5. There is a few analysis. The sensory analysis would greatly enrich this study.

Author Response

Please see the attachment and the revised manuscript

Round 2

Reviewer 1 Report

No further comments.

Reviewer 2 Report

I still insist that this paper was not properly designed and performed. When preparing a processed meat product, edible meat quality (color, meat tenderness, texture and flavor), especially consumer evaluation is needed. This paper has conducted three meat products including boneless belly, bacon slices, and sausage patties. None of the consumer rating has been done. In addition, the authors has not give compelling evidence for the difference of detected parameters as color or lipid oxidation. Though the ingredient of pig feeding has been added, I am not satisfied with responses from the authors. 

Reviewer 3 Report

The authors have well response to the reviewer's comments and revised the manuscripts. I have no further comments.